# OpenReview forum: "Understanding the Dilemma of Unlearning for Large Language Models"
_ICLR.cc/2026/Conference — ICLR 2026 Conference Withdrawn Submission_

### Official Review · Reviewer_JtCC · 2025-10-28

**Soundness:** 3
**Presentation:** 3
**Contribution:** 2
**Rating:** 4
**Confidence:** 4

**Summary:**

The paper introduces UNPact, an interpretable framework for analyzing unlearning in large language models (LLMs) using prompt attribution and token-level contribution tracking.
By quantifying the influence of individual prompt tokens, UNPact allows for the comparison of model behavior before and after unlearning, as well as the inspection of what is actually altered.
Through experiments on six unlearning methods, three LLMs, and three datasets, the paper evaluates why unlearning appears to work, whether knowledge is really erased or recoverable, and why catastrophic forgetting happens.
Key findings reveal that existing unlearning approaches often disrupt focus on salient tokens but fail to irreversibly erase knowledge.

**Strengths:**

- The proposed method is straightforward and easy to understand.
- The empirical design spans multiple unlearning methods, ensuring comprehensive evaluation.
- The proposed framework UNPACT is simple yet powerful, allowing interpretability analysis applicable to both open- and closed-source LLMs.

**Weaknesses:**

- Investigating the limitations of existing unlearning methods and the causes of catastrophic forgetting at the token level is not particularly novel. Previous studies [1,2] have already discussed these aspects in detail, which reduces the novelty of this paper.

- The proposed method involves many sensitive hyperparameters for binary comparisons, which may reduce its reliability in practical applications. Moreover, the paper does not provide detailed ablation studies to show how these hyperparameters affect performance.

- The framework relies heavily on perturbation-based token analysis, which may be computationally expensive and sensitive to prompt phrasing.

- Some metric definitions, such as the recovery rate and destructive rate, are vague and potentially confusing.


[1] Selective Forgetting: Advancing Machine Unlearning Techniques and Evaluation in Language Models. AAAI 2025.

[2] ReLearn: Unlearning via Learning for Large Language Models. ACL 2025.

**Questions:**

- In lines 342–344, the authors state that “we define the recovery rate as the proportion of supposedly forgotten knowledge that can be restored.” This definition is vague, and the paper does not provide a detailed explanation of how the recovery rate is actually computed. The same issue applies to the definition of the destructive rate later in the paper.

- I would like to know how the hyperparameters in Equations (3) and (4) are set, and whether the same hyperparameter values are applied to all samples in the unlearned set.

- In line 357, the authors mention that “explicitly emphasizing the KEYTOKENS in the prompt elevates forgotten knowledge to the model’s Top-1 prediction.” I understand that this may increase the prediction probability of the forgotten knowledge, but does it always reach the Top-1 prediction? I am also curious whether combining FOCUSONKEY with PROBAB would further improve recovery performance.

- The datasets used in the paper contain QA pairs with relatively short answers. Would the proposed method still perform well on more complex datasets with longer answers, such as the TOFU dataset?

---

### Official Review · Reviewer_SeeX · 2025-10-28

**Soundness:** 3
**Presentation:** 3
**Contribution:** 2
**Rating:** 4
**Confidence:** 2

**Summary:**

This paper introduce UNPACT, an interpretable framework for unlearning. It quantifies each prompt token’s influence on outputs, en- abling pre- and post-unlearning comparisons to reveal what changes. Experiments on six unlearning methods, three LLMs, and three benchmark show that existing methods tend either to be insufficient or destructive. However, the "UNPACT" framework proposed by the author is more empirical and lacks theoretical support.

**Strengths:**

- The experimental setup is complete. Detailed experiments are conducted on various model structures and multiple baselines.
- This paper shifts the research focus from studying “how to forget” to “the explainability of forgetting”.
- The conclusions of the paper provide a clear research direction for the future.

**Weaknesses:**

- Although this paper summarizes and implements the existing methods, it does not provide corresponding solutions based on the phenomena obtained.
- Computing UNPACT requires multiple forward passes through the model, which is computationally expensive (especially for long sentences), affecting the usefulness of UNPACT as a diagnostic tool.
- Is there an unlearning explanation for the MoE model?
- Although the authors proposed the "UNPACT" framework to explain the unlearning mechanism, the entire approach is more empirical and heuristic, without providing theoretical support or mathematical proof.

**Questions:**

Can you design a corresponding unlearning solution based on the conclusions you draw?

---

### Official Review · Reviewer_4WGL · 2025-10-30

**Soundness:** 2
**Presentation:** 3
**Contribution:** 2
**Rating:** 2
**Confidence:** 2

**Summary:**

This paper aims to reveal and understand the dilemma of unlearning for large language models. Specifically, it proposes an interpretable framework via prompt attribution and contribution tracking, and discusses three questions related to unlearning, e.g., why unlearning can work, whether knowledge is really unlearned, and why catastrophic forgetting happens, through auditing what changes before and after prompt masking.

**Strengths:**

1. This paper focuses on the LLM unlearning from the view of interpretability, which is important and of high significance to provide insights for future unlearning paradigm or other design.
2. The visualization is great to illustrate the specific findings and observations regarding the designed mechanism.
3. Several representative unlearning methods are considered in the experimental part to support the empirical findings and claims.

**Weaknesses:**

I appreciate the authors' idea and presentation for the prompt attribution and contribution tracking in LLM unlearning, while I still have concerns and questions for the current version, which may be considered to enhance the overall quality and the rationality of the claim:
1. Some of the presentation claims are not accurate and faithful in summarizing the current literature and questionable; please find specific questions for further discussion and revision.
2. Although the paper presents illustrative examples (e.g., key-token analysis), it is questionable whether these examples provide generalizable or faithful insights into the unlearning mechanism. And some definitions can be further elaborated and explained.
3. The experiments appear preliminary, as they cover a limited set of models and unlearning methods.

**Questions:**

1. I'm concerned with the claim of "interpretability analysis of the mechanism is scarce due to the difficulty of tracing knowledge in LLMs' complex architectures", as it didn't cover a series of works on knowledge editing in LLMs, in which area there is also some methods capable of interpretability for analyzing the what changes before and after post-hoc adjustment of LLMs.
2. The former also induces a critical question about the uniqueness of the contribution, whether it is closely related to the unlearning scenario. If not, what is the major contribution if the authors adopt some prior interpretability works to analyze the problem under LLM unlearning.
3. There are already a series of work in unlearning and knowledge editing answers the similar question as highlighted in the paper, e.g., "why unlearning can work" (works like every conventional unlearning methods will analyze the working mechanism), "Is the knowledge really unlearned" (some previous work also explores and reveal the knowledge is not truly deleted and recoverable), "why catastrophic forgetting happens" (a lot of unlearning works focuses on the trade-off will also target the problem). Please consider to do a more sufficient and comprehensive literature survey to better position the current presentation, and highlight the unique contribution and different findings.
4. Although the authors have shown a lot of great and beautiful examples of keytokens to try to answer the question of when target knowledge is forgotten, it is questionable whether the mechanism can realize the general or faithful insights for better understand the unlearning effects.
5. In different unlearning settings or scenarios, there would be diverse and flexible definitions for achieving satisfactory unlearning. I'm curious what is the "gap to reliable unlearning", and how is "reliable unlearning" is defined in the claims related to the dilemma of unlearning illustration? As it seems to be an important concept, but need further. explanation.
6. The experimental results didn't cover different models and more advanced unlearning methods, which seems to be preliminary and not mature to draw conclusions.

---

### Official Review · Reviewer_kMMR · 2025-11-01

**Soundness:** 2
**Presentation:** 2
**Contribution:** 2
**Rating:** 2
**Confidence:** 3

**Summary:**

This paper investigates the dilemma of unlearning in LLMs: the tension between insufficient forgetting (where “forgotten” knowledge can be recovered) and catastrophic forgetting (where general capabilities collapse).

The authors propose UNPACT, an interpretable framework for analyzing unlearning from the prompt perspective. UNPACT measures each prompt token’s contribution to model outputs (via log-probability differences), identifies KEYTOKENS, and compares them before and after unlearning to interpret how attention shifts.

Experiments are conducted on four training-based unlearning methods, three LLMs, and three datasets (News, Books, WMDP).

Key findings include:

- unlearning mainly disrupts focus on key prompt tokens;
- forgotten knowledge can be easily recovered by emphasizing those keywords;
- catastrophic forgetting arises from indiscriminate penalization of all tokens.

**Strengths:**

1. UNPACT provides an interesting angle, applicable to both open- and closed-source LLMs.
2. The authors systematically study four methods, multiple models, and datasets, offering a broad empirical view.
3. The work highlights important limitations of existing unlearning methods and articulates the inherent trade-off problem.

**Weaknesses:**

1. Limited methodological novelty: UNPACT is primarily an application of standard token attribution (perturb-and-measure log-prob difference). The conceptual novelty beyond existing saliency or prompt-influence methods (e.g., Captum) is minimal. How does UNPACT differ technically from prior token-level attribution or saliency approaches?
2. Results are only on small/medium models (smaller than 14B). Claims about LLM-level unlearning generality are thus not well-supported for a pure analysis paper. More models and families should be considered to validate the analysis.
3. “Recovery rate” and “destructive rate” are not precisely formalized or statistically analyzed.
4. The paper lacks quantitative or conceptual comparison with existing attribution or attention-based interpretability techniques, making its unique contribution unclear.
5. Preliminary and incomplete contribution: The paper reads more like an early-stage exploratory study of a potential key-token-based unlearning method, rather than a complete piece of research. While the observations are interesting, the work stops short of demonstrating how these insights can be effectively utilized to improve unlearning methods. As a result, the contribution feels preliminary and insufficient to meet the standard of a mature paper.

**Questions:**

Please see the weakness above. And also:

1. How is GPT-4o-mini’s judgment validated against human or metric-based evaluation? Why using GPT-4o-mini only?
2. On the FOCUSONKEY recovery experiment (Tabel 2): Could emphasizing KEYTOKENS alter the task itself rather than recover forgotten knowledge? How do you ensure that recovery is not prompt leakage or external information injection?
3. Is UNPACT intended as a new unlearning method or an interpretation tool? The current positioning oscillates between the two.
4. Does the interaction among phrases or segments of tokens also matter? The multi-token dependencies might better reflect contextual semantics.

---

### Note · Authors · 2025-11-17

I have read and agree with the venue's withdrawal policy on behalf of myself and my co-authors.